# A Study on Nanoleakage of Apical Retrograde Filling of Premixed Calcium Silicate-Based Cement Using a Lid Technique

**DOI:** 10.3390/ma17102366

**Published:** 2024-05-15

**Authors:** Nyamsuren Enkhbileg, Jin Woo Kim, Seok Woo Chang, Se-Hee Park, Kyung Mo Cho, Yoon Lee

**Affiliations:** 1Department of Periodontics and Endodontics, School of Dentistry, Mongolian National University of Medical Sciences, Ulaanbaatar 14210, Mongolia; naya.nyamsuren@gmail.com; 2Department of Conservative Dentistry, College of Dentistry, Gangneung-Wonju National University, 7 Jukheon-gil, Gangneung 25457, Republic of Korea; mendo7@gwnu.ac.kr (J.W.K.); drendo@gwnu.ac.kr (S.-H.P.); drbozon@gwnu.ac.kr (K.M.C.); 3Department of Conservative Dentistry, School of Dentistry, Kyung Hee University, Seoul 02447, Republic of Korea; swc2007smc@khu.ac.kr

**Keywords:** calcium silicate-based putty, mineral trioxide aggregate, nanoleakage, retrograde filling, sealing ability, lid technique

## Abstract

This study aimed to compare the nanoleakage of retrograde fillings with premixed calcium silicate-based putty and mineral trioxide aggregate (MTA), using two different techniques (traditional and Lid). Sixty-four extracted human teeth were decoronated, then root canals and ends were instrumented for retrograde filling and divided into four groups according to the retrograde filling technique: the traditional and the Lid technique. Each group (n = 15) was filled with Ceraseal + Well-Root putty, Well-Root putty, Ceraseal + ProRoot MTA, and ProRoot MTA. The nanoleakage was evaluated using the Nanoflow device (IB Systems) on days 1, 3, 7, 15 and 30. Data were collected twice per second at the nanoscale (nL/s) and calculated after archiving the stabilization of fluid flow. The Kruskal–Wallis and Mann–Whitney U-tests were used for statistical analysis. All groups showed enhanced sealing ability over time. Regardless of filling materials, the Well-Root putty, Ceraseal+Well-Root putty, and Ceraseal+ProRoot MTA groups indicated less nanoleakage than the ProRoot MTA group in the first week of evaluation (*p* < 0.05). Although all groups did not show significant differences after 2 weeks, the Ceraseal+ProRoot MTA group leaked less than ProRoot MTA on Days 3 and 7 (*p* < 0.05). The scanning electron microscopic examined good adaptation to the cavity wall, which was similar to nanoleakage results. Premixed calcium silicate-based putty retrograde filling material alone and using the “lid technique” were shown to be faster and less prone to nanoleakage when compared to MTA.

## 1. Introduction

Periapical surgery is performed in the presence of persistent peri-radicular lesions when non-surgical treatment is impossible and/or failed [1]. A meta-analysis of clinical studies indicates that long-term endodontic microsurgical operations have a 91.3% success rate and a 79–100% survival rate [2]. Microsurgery and novel root-end filling materials have high biocompatibility and the ability to create an efficient barrier between the retrograde filling material and the periapical tissues. The success of the microsurgery depends on using proven retrograde filling materials, which are required to adhere to the tooth structure, be insoluble in tissue fluids, be dimensionally stable, nonresorbable, and radiopaque, and exhibit biocompatibility [3]. Currently, mineral trioxide aggregate (MTA) fulfills almost all the requirements of an ideal retrograde filling material that is commonly used [4]. However, MTA has a long setting time, discoloration and difficult handling characteristics, which facilitates leakage and prolongs the contact time of filling material with vital tissues [5,6]. To address some of the handling issues, calcium silicate-based pastes and putty materials have been developed. Once set, these materials have properties comparable to MTA, allowing for the flow of the premixed substrate with sufficient working consistency while reducing the setting time [7,8,9]. Premixed putty types of calcium silicate-based materials have also been marketed as an injectable Root Repair Filling Material Putty, Endosequence Root Repair Material, TotalFill BC RRM Putty, and Well-Root putty (WR). These materials have excellent mechanical and biological characteristics with good handling properties [10].

Some clinicians recommend an updated retrograde filling technique during periapical surgery. Dr. Nasseh created the revised retrograde filling technique known as the “Lid Technique” to facilitate the filling process. The Lid technique was described and first shared in 2008 after developing a premixed putty type of calcium silicate-based cement. The Lid technique is a combination of the light body effects of syringeable former material followed by placing a lid of the putty to protect this material during the setting period [11]. This technique is simple to use, reduces procedure time and prevents voids during application, which improves the retrograde filling quality [12].

The sealing ability of retrograde filling materials was achieved by several leakage studies using various methodologies such as dye penetration, bacterial penetration, radioisotope penetration, the electrochemical method and the fluid filtration method [13,14,15]. The clinical relevance of some leakage measurement methodologies is debatable because the results of these in vitro studies do not correlate with clinical outcomes, so such research is no longer generally accepted in the mainstream endodontic literature [16]. A nanoflow sub-nanoliter scaled fluid flow measuring device (NFMD) obtains precise and highly reliable test results by detecting fluid flow with a light-sensitive photodiode with a resolution of up to 0.196 nL. NFMD is measured in dentinal fluid flow at the level of nanoleakage (nL/s) of adhesives or cements [17]. Likewise, if measurement and comparison of the leakage of retrograde filling could be collected at the nanoscale, it would be possible to provide solid results on the sealing ability of retrograde filling materials and techniques. However, to the best of our knowledge, no studies comparing the nanoleakage of root-end filling materials are available. Therefore, the aim of this study was to investigate and compare the nanoleakage of premixed calcium silicate-based Putty and MTA, using the traditional and Lid techniques over a time up to 30 days. The null hypothesis was that no difference exists between the two materials and retrograde filling techniques regarding their sealing ability.

## 2. Materials and Methods

### 2.1. Sample Preparation

This study protocol was approved by the Institutional Review Board (GWNU-IRB2021-A009). We collected 60 human single-rooted premolars extracted for orthodontic reasons that met the following criteria: straight root with a single root canal, no visible root caries, fractures, or cracks, and a completely formed apex on visual examination. (Figure 1). Sample size was determined according to previous studies with similar methodology [18,19,20,21,22]. All specimens’ external root surfaces were cleaned by ultrasonic instruments and placed in normal saline to keep them moist before the experiment. The crown was removed from each tooth and roots were trimmed to standardize the root canal length at 10 mm to reduce the possibility of errors arising from differences in root canal length [14]. Each root canal was instrumented to a size of #40 with a 04 tapered Protaper Universal rotary file (Dentsply Sirona, York, PA, USA) and irrigated with 1.0 mL of 2.5% NaOCl alternating with 17% EDTA between instruments. Three millimeter of the root-end was removed and an apical preparation 3 mm depth was performed using KiS-2D Retroprep tips (Spartan Obtura, Algonquin, IL, USA) in the EIE ultrasonic system. A size 40/04 gutta-percha (GP) point was inserted into each canal, and the extruded tip was trimmed evenly with the root tip. The GP point was then removed, the apical 3 mm sectioned, and the point was reinserted into the canal to serve as a stop for placement of the root-end fillings [14,23]. All prepared specimens were randomly allocated and numbered to the following four groups (n = 15/each) [20,22,24]. Each group of 15 roots received a retrograde fill of testing Ceraseal (META BIOMED, Cheongju, Republic of Korea), Well-Root putty PT (Vericom, Chuncheon, Republic of Korea), and ProRoot MTA (Dentsply International, Johnson City, TN, USA) materials by the corresponding technique (Table 1).

The materials were used according to the manufacturer’s directions. The GP point was then removed to confirm that the leakage was caused solely by the apical filling material. Each root was prepared for nanoflow testing as follows: a 0.9 mm diameter and 10 mm long metal tube was inserted at the center of transparent 10 × 10 × 2 mm Plexiglas film and sealed with All Bond Universal [17] (Bisco Inc, Schaumburg, IL, USA). The root was affixed to the other side of the film and a metal tube was inserted into the cervical orifice of the root canal to a 2 mm depth. The spaces between the film and root dentin were filled with a flowable composite resin (G-aenial Flo, GC, Tokyo, Japan). All surfaces of the roots were covered with nail varnish twice except for the retrograde filled apical area to prevent undesirable leakage (Figure 2A) [19]. Prepared roots were wrapped with moist gauze and stored in a chamber with 100% humidity at 37 °C [25].

### 2.2. Evaluation of Nanoleakage

The inserted metal tube of each prepared specimen was connected to a glass capillary by silicone tubing filled with distilled water from an NFMD [17,19,21]. A distilled water-filled glass capillary with an internal diameter of 0.5 mm was connected between a water reservoir and the specimen. An air bubble was created into the capillary, through which the flow of distilled water could be detected by using a photosensor. The movement of the air bubble and the volume of moved distilled water were measured and automatically recorded by the computer software (Figure 2B) [17].

The flow rate was measured for 600 s at a pressure of 50 cm of H_2_O at 21 °C, and the stabilization time of the flow rate was achieved after 5 min [19]. Measurements of nanoleakage were made at 1, 3, 7, 15, and 30 days after the placement of the experimental materials [19,20,21,25,26,27,28].

### 2.3. Scanning Electron Microscopy (SEM) Examination

After the leakage test, four specimens in each group were randomly selected and subjected to SEM examination. The root was sectioned perpendicular to its long axis to obtain a section of 1.5 in thickness using a high-speed handpiece for longitudinal cross-section images. The specimens were etched with 37% phosphoric acid for 15 s and rinsed, and then the specimens were dried according to the protocol suggested by Perdigao et al. [29]. All specimens were platinum-coated before observation under the SEM (Hitachi S-4700; Hitachi, Tokyo, Japan). And the interfaces between retrograde filling materials and dentin were observed. 

### 2.4. Statistical Analysis

Statistical analysis was performed using SPSS ver. 26 software (SPSS Inc., Chicago, IL, USA). MANOVAs with Pillai’s Trace and Wilks’ Lambda test were used to statistically analyze the differences and mean values of nanoleakage. A Kruskal–Wallis nonparametric test was used to analyze differences in nanoleakages across groups. The Mann–Whitney U-test was used for pair-wise comparisons. The selected level of significance was ρ < 0.05.

## 3. Results

### 3.1. Nanoleakage Measurements Using NFMD

The nanoleakage measurement was determined after stabilization of fluid flow from 300–600 s. Table 2 shows the nanoflow rate of each group, according to the days of evaluation and statistical result.

The nanoleakage of all groups was significantly higher on Day 1 than on the other evaluation days. However, the reduction in nanoleakage in the C + WR, WR only, and C + PR groups began on Day 3, while the PR-only group began on Day 15 (ρ < 0.05). Upon comparing all groups among the time intervals, the results showed statistically significant differences in nanoleakage between groups on Days 3 and 7 (Figure 3).

### 3.2. SEM Examination

All materials exhibited acceptable adhesion to the dentin wall. The Lid Technique specimens had fewer gap points and no remarkable separate phase between two materials. However, in the C+PR group specimens, the differences between these two materials were more noticeable. There were some cracking and gap points could be detected in all groups (Figure 4).

## 4. Discussion

One of the key factors for the success of root-end filling material is its ability to achieve a good fluid tight seal at the apex, which inhibits penetration of bacteria and bacterial products from the root canal system into the periapical tissues [30]. Furthermore, the retrograde filling technique, material insertion, compaction, barrier thickness, and evaluation time can all influence the sealing ability of apical barriers [2]. Hence, this study evaluated the sealing ability of a premixed type of calcium silicate-based putty and MTA and compared two different retrograde filling techniques according to time.

A previous research project evaluated the bacterial leakage of MTA over three time periods and found that 60–70% of specimens were sealed after 10 days, and then sealing ability reached 100% at 30 days [31]. This study investigated the nanoleakage of retrograde materials over five timepoints: on the 1st, 3rd, 7th, 15th, and 30th days. The results showed that time influences the sealing ability of retrograde filling materials: the WR-only-group nanoleakage was reduced on the third day of investigation, and that of the PR-only group was significantly reduced on Day 15. However, there was no statistically significant difference in nanoleakage on Day 15 and 30. The results of this study were in line with those reported by Karobari MI et al. that microleakage of MTA was not statistically different between 15 and 30 days of evaluation [18]. Also, several studies found that MTA had no microbial growth after 30 days [25,27].

We used an NFMD, which measures the amount of fluid up to the nanoliter/second unit in real-time without destroying the tooth. This equipment can provide extremely precise information about the material’s sealing ability [21]. In addition, after flow stabilization, a flow rate analysis was performed to allow for quantitative and qualitative analyses of nanoleakage.

A few studies used NFMD to investigate the nanoleakage of canal filling materials [19,21]. To our knowledge, the nanoleakage of premixed calcium silicate-based retrograde filling materials have been never evaluated according to filling technique and time. The nanoleakage test showed the premixed type of calcium silicate-based putty had significantly less leakage than MTA in the first week of evaluation. Similar to the present study, related studies have found that premixed calcium silicate-based putty materials have a better sealing ability than MTA [20,32]. Some studies have shown that the bacterial leakage of two materials have similar sealing abilities [23,26,33]. Contrary to this, Hirschberg, Craig S et al. noted that the sealing ability of premixed putty samples leaked substantially more than MTA samples [34]. These differences could be due to the different study designs and materials used in each study.

More than ten years of clinical experience demonstrated that the Lid technique demonstrated similar or better results than the traditional retrograde filling technique [11]. Rencher B et al. compared the bacterial leakage of retrograde bioceramic putty alone and in combination with bioceramic paste. After 30 days of incubation, the results revealed that bioceramic putty had a better sealing ability than MTA. However, no statistically significant differences were found between bioceramic putty, BC putty plus paste, and MTA [14]. Our results demonstrated that on the third and seventh days, specimens placed using the Lid method had statistically significantly better sealing ability than ProRoot MTA alone (ρ > 0.05). This could be attributed to the texture, long setting time, and poor handling properties of MTA. Mixing MTA with sterile water forms a sandy mixture that is difficult to deliver to the operative site and to condense properly [35]. The decreasing setting time improved the working properties of MTA [36,37]. A short setting time and application periods, as well as convenience application, are additional benefits of premixed calcium silicate-based root-end filling materials, which are especially important in clinical outcomes [28]. The Lid technique streamlines the retrograde filling procedure and may improve the efficiency of root-end obturation when using a combination of sealer and retrograde filling materials.

The aim of the SEM examination was focused on observing the connection and characteristics of the gaps between experimental materials and their adaptation to the dentin wall surface. The SEM analyses were similar to nanoleakage data; groups with superior sealing abilities had better adaptation to the dentin wall than groups with inferior sealing abilities. The combination of Ceraseal and Well-Root putty materials was more blended and had fewer differences than MTA, which could be attributed to the material type and structure. The particle size was reduced to nano and microparticles to improve the clinical handling properties of the new premixed calcium silicate-based cement materials [38]. As a result, the particle size of premixed type materials may influence the connection and differentiation of Ceraseal and Well-Root putty materials in SEM images. Nevertheless, cracks and gaps/voids between experimental materials and dentin were evident during specimen preparation, even following the Perdigao et al. specimen dehydration protocol [29]. Further research is needed to confirm this finding.

This study had some limitations, including the possibility of the differentiation of canal anatomical conformation, which would increase the fluid flow through the samples. Furthermore, an interfacial gap or crack between retrograde materials and dentin during the preparation and desiccation of SEM specimens. Assessing the gap size of samples may have yielded far more detailed findings, confirming the nanoleakage measurements. We tested only one product of premixed calcium silicate-based sealer and putty materials. Accordingly, future research with various products, including gap size evaluation, is required. Furthermore, the outcomes of in vitro studies cannot be immediately extrapolated to clinical conditions, necessitating further clinical trials.

## 5. Conclusions

Within the limitations of the present study, our findings indicate that the premixed calcium silicate-based retrograde filling putty material leaked faster and had lower nanoleakage. The Lid technique had a better sealing ability than MTA alone, indicating that it might be employed successfully and efficiently during endodontic microsurgical treatments.

## Figures and Tables

**Figure 1 materials-17-02366-f001:**
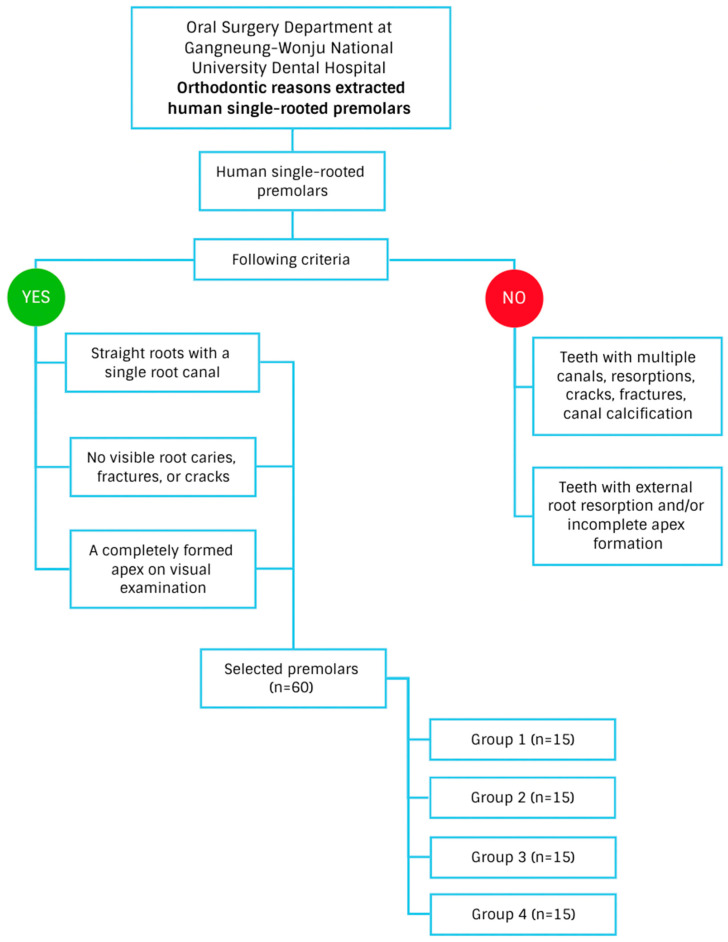
Flowchart of the tooth sample collection.

**Figure 2 materials-17-02366-f002:**
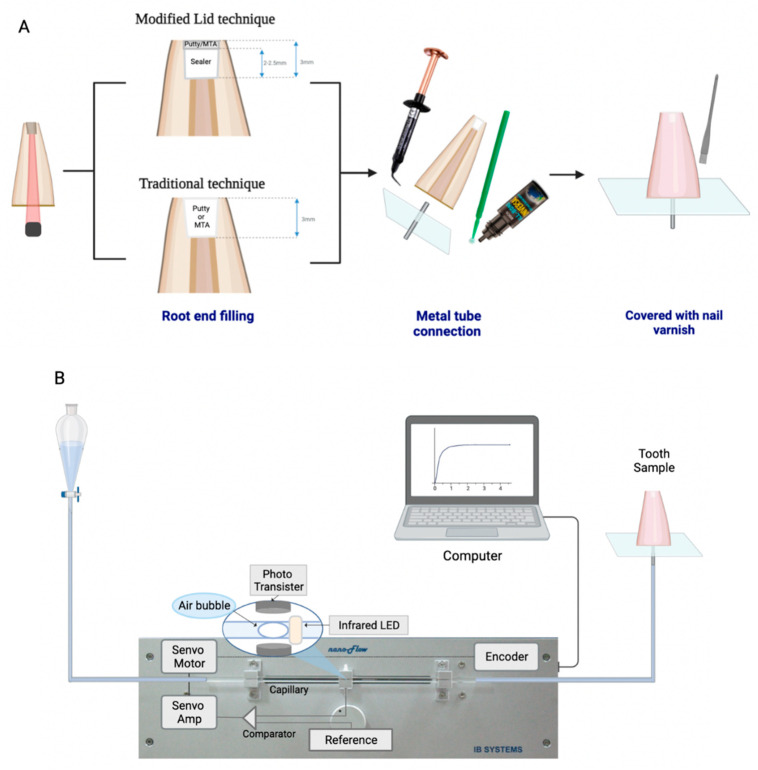
A schematic diagram of the tooth sample and testing device. (**A**) A tooth sample preparation. (**B**) Nanoflow subnanoliter-scaled dentinal tubular fluid flow measurement device (NFMD) connected to a specimen.

**Figure 3 materials-17-02366-f003:**
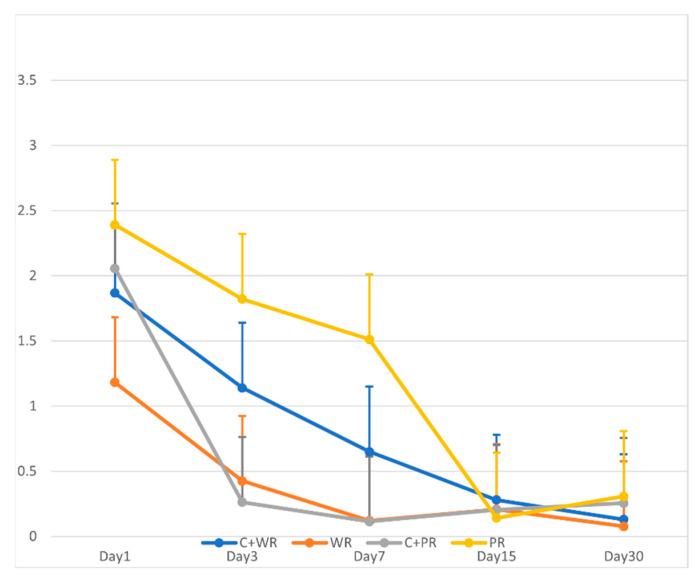
Line chart of the mean and SD nanoflow rate comparison of all experimental groups: C + WR Ceraseal + Well-Root PT; WR Well-Root PT only; C + PR Ceraseal + ProRoot MTA; PR ProRoot MTA only.

**Figure 4 materials-17-02366-f004:**
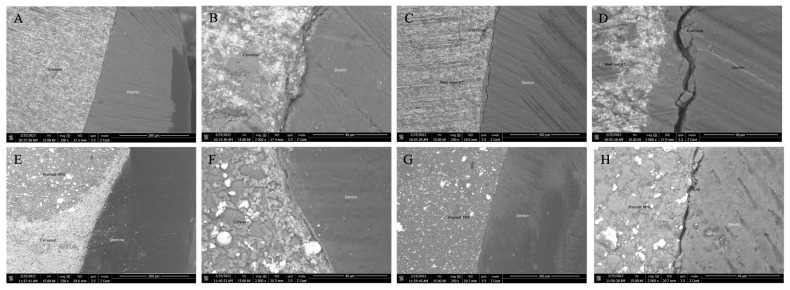
Scanning electron micrograph on longitudinal cross-section of root-end filled specimens. (**A**) Ceraseal + Well-Root PT group (×250). (**B**) Ceraseal + Well-Root PT group (×2000). (**C**) Well-Root PT group (×250). (**D**) Well-Root PT group (×2000). (**E**) Ceraseal + ProRoot MTA group (×250). (**F**) Ceraseal + ProRoot MTA group (×2000). (**G**) ProRoot MTAgroup (×250). (**H**) ProRoot MTA group (×2000).

**Table 1 materials-17-02366-t001:** Experimental setup and groups.

Groups		Number of Specimens	Retro-Filling Method
Group 1	Ceraseal+ Well-Root putty (C + WR)	15	root-end cavity filled with calcium-silicate based sealing material and the putty lid was placed.
Group 2	Well-Root putty (WR) only	15	root-end cavity filled only with calcium-silicate based putty material
Group3	Ceraseal+ ProRoot MTA(C + PR)	15	root-end cavity filled with calcium-silicate based sealing material and placing lid of MTA
Group 4	ProRoot MTA (PR) only	15	root-end cavity filled only with MTA

**Table 2 materials-17-02366-t002:** Distribution of nanoleakage difference, according to retrofilling material and day of evaluation.

Time	N	C + WR		WR		C + PR		PR		*p*-Value
		Mean	SD	Mean	SD	Mean	SD	Mean	SD	
Day1	60	1.867	2.036	1.182	0.954	2.056	1.987	2.390	2.651	0.836
Day3	60	1.137	1.378	0.426	0.382	0.262	0.404	1.822	3.008	0.006
Day7	60	0.645	0.881	0.122	0.144	0.115	0.086	1.512	2.068	<0.001
Day15	60	0.276	0.398	0.208	0.377	0.204	0.162	0.142	0.219	0.500
Day30	60	0.125	0.150	0.077	0.100	0.255	0.232	0.307	0.546	0.195

C + WR Ceraseal + Well-Root putty; WR Well-Root putty only; C + PR Ceraseal + ProRoot MTA; PR ProRoot MTA only.

## Data Availability

The datasets generated during and/or analyzed during the current study are not publicly available due to institutional policy but are available from the corresponding author on reasonable request.

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
