# Peer review of "A Study on Nanoleakage of Apical Retrograde Filling of Premixed Calcium Silicate-Based Cement Using a Lid Technique"

_materials, 2024, doi:10.3390/ma17102366_

Round 1
Reviewer 1 Report
Comments and Suggestions for Authors
The article is potentially interesting, some minor revisions are needed, as well as English editing.
Abstract.
Please give manufacturer and location in brackets for Ceraseal and Well-Root.
Please add manufacturer's location to IB Systems.
Please use either Well-Root PT, or Well-Root putty, or Well-Root, please check the entire text, not only the abstract.
Introduction.
Line 54-58, 62-63, 76-77. Please check, something is not right.
Mat and Meth.
Line 103-106. Please be more specific, which product is MTA based, which is calcium-silicate based.
The Results and Discussion are well done.
Please display conclusion as a separate, numbered section. The conclusion should be more detailed.
Comments on the Quality of English Language
needs improvement
Reviewer 2 Report
Comments and Suggestions for Authors
The article is well-developed and interesting. I found it engaging to read. However, there are a few areas for improvement. For instance, in Figure 2, it would be helpful to include not only the mean of the filtrations but also the standard deviations.
The discussion should address the various protocols' inherent aspects, such as complications, increase in steps, cost, and influence on clinical outcomes.
My primary criticism is that the article is mostly an in vitro study with a focus on clinical practice. To make it more relevant, the study could incorporate some novel elements about the materials used from a material science perspective. Alternatively, the article could be published in a journal like Dentistry.
Reviewer 3 Report
Comments and Suggestions for Authors
The aim study was to investigate and compare the nano leakage of premixed calcium silicate-based Putty and MTA, using traditional and Lid techniques over time up to 30 days. The aim of this paper is quite interesting. This research is under the scope of this journal; the topic is relevant for readers and this research deals with potentially significant knowledge to the field.
However, some aspects are possibly improved in the manuscript:
(Keywords)
- Please order the keywords/Mesh terms alphabetically
Introduction
- Endodontic microsurgery is often the last option when nonsurgical retreatment fails, is unfeasible or is unlikely to improve the initial endodontic treatment. In particular, only surgical intervention may resolve cases involving a persistent lesion (microbial infection) with etiology related to complex canal anatomy (https://doi.org/10.3390/biomedicines8100383). This study opens new ways of endodontic planning and future studies in endodontic root canal treatment and microendodontic surgery. In this last one also in the retropreparion field, one is the selection of the tip size (https://doi.org/10.3390/biomedicines8100383). Recent Systematic Review and Meta-analysis (Long-Term Prognosis of Endodontic Microsurgery) showed high success rates and predictable results can be expected when EMS is performed by trained endodontists, allowing good prognosis and preservation of teeth affected by secondary AP. A pooled proportion of success rate of 91.3%, from an overall amount of 453 treated teeth included in RCT; from an overall 839 included teeth in PCS, a pooled success rate of 78.4% was observed, with the follow-up time ranging from 2 to 13 years. Survival rate outcomes varied from 79% to 100% for the same follow-up period. Five prognostic factors influencing the outcome were disclosed: smoking habits, tooth location and type, absence/presence of dentinal defects, interproximal bone level, and root-end filling material. Please, read the study were high level of clinical evidence has been produced highlighting the influence the studied biomechanical factors on the prognosis, specifically systematic reviews and meta-analysis, but the impact of each factor on the overall outcome remains unclear.
(Statement of Clinical Relevance)
- Identified the aim and null hypothesis on the end of the introduction.
(M&M)
- Major: This experimental procedure does represent the normal clinical procedure of microendodontic surgery. Why did the retroprepation before the canal obturation?
- What was the magnification used to perform the experimental procedure?
- When mentioning materials or devices: for some of them, Authors don't mention the manufacturer at all, for some you mention only the manufacturer, for some the manufacturer and city, for some you mention the manufacturer and city/ country.
- How was the sample calculated? Did the authors perform a power analysis to evaluate if this sample size was appropriate? How did you divide the teeth in diferents groups? And did it by randomized?
- How many operators performed the experiments?
- It is recommend add a flowchart, to explain to reads the sequence of the study and N.
- Which results are comparable with other articles? What has this study been new? Paulo et al, https://doi.org/10.3390/app11156849 evaluate the effect of blood contamination (similar of our study) on the push-out bond strength obtained with three different biomaterials to root canal dentin refer in their study. Overall results indicate TotalFill presents the highest push-out bond strength values, followed by Biodentine and, lastly, MTA. Blood contamination did not affect the dislodgement resistance. Biomaterials’ comparison within each radicular segment revealed both Biodentine as TotalFill and as the preferable alternatives for application in the coronal region. TotalFill might be the biomaterial of choice for placement in the apical region.
(Discussion)
- Micro versus Nano leakage.
- “Literature suggests some of the in vitro conditions in which several studies were performed as a possible reason for the formation of microcracks: stresses exerted during extraction (either traumatic or atraumatic extraction techniques), possible tooth dehydration, inappropriate storage and careless handling of the extracted teeth, as well as absence of periodontal support. Therefore, in the present ex vivo study we could have obtained an overestimation of microcracks, despite all the efforts that have been made to prevent it—only freshly extracted teeth were included and specimens were kept moist throughout the experimental procedures, along with the use of silicon blocks to minimize the concern of the absence of periodontal ligament and stabilize the teeth during instrumentation procedures, although in Gondim et al. , the use of stabilization methods did not prevent the appearance of fractures after root-end preparation. In order to avoid artifacts and obtain results which are more clinically relevant, some authors claim that investigations should be preferably performed in situ. Calzonetti suggested that, in situ, roots may absorb some of the ultrasonic impacts and prevent microcracks propagation, overcoming tooth dehydration and brittleness associated with in vitro context, thus reducing the chance of artefacts. Previous studies indicate the use of cadavers as a potentially suitable alternative”
- Please, identified what were the strength(s) and more limitations of this study (as diferentes canal conformation )? And also, implications for future perspectives.
Reviewer 4 Report
Comments and Suggestions for Authors
The Authors must see my remarks

Author Response
The reviewer suggested throughout the manuscript to add more references to back the methodology. We have revised the manuscript accordingly.
The authors thank the reviewer for the constructive comments.
Round 2
Reviewer 1 Report
Comments and Suggestions for Authors
I have no further comments. Thank you for the revisions.
Reviewer 2 Report
Comments and Suggestions for Authors
The authors have appropriately addressed the suggested changes.
Reviewer 3 Report
Comments and Suggestions for Authors
the authors improve the article